# Low-Energy Shock Wave Suppresses Prostatic Pain and Inflammation by Modulating Mitochondrial Dynamics Regulators on a Carrageenan-Induced Prostatitis Model in Rats

**DOI:** 10.3390/ijms24043898

**Published:** 2023-02-15

**Authors:** Zong-Sheng Wu, Hung-Jen Wang, Wei-Chia Lee, Hou Lun Luo, Tsu-Kung Lin, Yao-Chi Chuang

**Affiliations:** 1Department of Urology, Kaohsiung Chang Gung Memorial Hospital, College of Medicine, Chang Gung University, Kaohsiung 833, Taiwan; 2Center for Shockwave Medicine and Tissue Engineering, Kaohsiung Chang Gung Memorial Hospital, College of Medicine, Chang Gung University, Kaohsiung 833, Taiwan; 3School of Medicine, College of Medicine, National Sun Yat-sen University, Kaohsiung 833, Taiwan; 4Center for Mitochondrial Research and Medicine, Department of Neurology, Kaohsiung Chang Gung Memorial Hospital, Kaohsiung 833, Taiwan

**Keywords:** shock wave, chronic prostatitis, mitochondrial dynamics, carrageenan

## Abstract

A low-energy shock wave (LESW) has therapeutic effects on chronic prostatitis/chronic pelvic pain syndrome (CP/CPPS); however, its mechanism of action remains unclear. We explored the effects of LESW on the prostate and mitochondrial dynamics regulators in a rat model of carrageenan-induced prostatitis. The imbalance of mitochondrial dynamics regulators may affect the inflammatory process and molecules and contribute to CP/CPPS. Male Sprague–Dawley rats received intraprostatic 3% or 5% carrageenan injections. The 5% carrageenan group also received LESW treatment at 24 h, 7 days, and 8 days. Pain behavior was evaluated at baseline, 1 week, and 2 weeks after a saline or carrageenan injection. The bladder and the prostate were harvested for immunohistochemistry and quantitative reverse-transcription polymerase chain reaction analysis. Intraprostatic carrageenan injection induced inflammatory reaction in the prostate and the bladder, decreased the pain threshold, and resulted in the upregulation of Drp-1, MFN-2, NLRP3 (mitochondrial integrity markers), substance P, and CGRP-RCP, whose effects were maintained for 1–2 weeks. LESW treatment suppressed carrageenan-induced prostatic pain, inflammatory reaction, mitochondrial integrity markers, and expression of sensory molecules. These findings support a link between the anti-neuroinflammatory effects of LESW in CP/CPPS and the reversal of cellular perturbations caused by imbalances in mitochondrial dynamics in the prostate.

## 1. Introduction

Chronic prostatitis/chronic pelvic pain syndrome (CP/CPPS), classified as either category IIIA or IIIB in the National Institutes of Health classification of prostatitis without bacterial infection, has been estimated to occur in up to 90% of patients with prostatitis [1]. Patients with CP/CPPS have more than 3 months of pelvic pain symptoms, in which some symptoms and discomfort are often related to the urinary bladder without bacterial infection [2]. Igarashi et al. reported that nerve growth factor, locally expressed in the bladder, is an important mediator inducing bladder overactivity with the upregulation of C-fiber afferent markers following 5% formalin-induced prostate inflammation in a rat model. Thus, the cross-sensitization between the prostate and the bladder causes the complexity of CP/CPPS symptoms [3]. Persistent pain not only leads to physical discomfort but also affects the psychological state and prevents people from working, thus placing a financial burden on patients and society [4].

Conventional therapies (such as antibiotics, anti-inflammatory drugs, and alpha-blockers) and other alternative therapies (such as phytotherapy and physiotherapy) have been used clinically [2,5]; however, these therapies did not have strong evidence to support their therapeutic effects. Therefore, there is an emergent need for a novel treatment.

Low-energy shock wave (LESW) therapy is non-invasive and widely used in clinical applications for various disorders. Therapeutic mechanisms of LESW include tissue repair, angiogenesis, anti-inflammation, nerve regeneration, and recruitment of stem cells [6,7,8]. LESW has been proven to demonstrate therapeutic effects on patients with CP/CPPS [9,10]. However, the mechanism of LESW for the treatment of CP/CPPS remains unclear. Previous studies have suggested that possible mechanisms include muscle tone reduction, hyperstimulation of nociceptors followed by desensitization, interruption of the flow of nerve impulses, and involvement of the neuroplasticity of pain memory [11]. Furthermore, Meng et al. quantitatively measured the change of plasma metabolites, and the results suggested that the glycine and serine metabolism pathways were involved in the LESW treatment in 75 CP/CPPS patients.

Mitochondria are intracellular organelles responsible for the production of the majority of adenosine triphosphate to provide cellular energy. In addition, mitochondria are known to participate in the regulation of ion homeostasis, programmed cell death, and reactive oxygen species (ROS) generation [12]. Mitochondrial dysfunction leads to the production of ROS and inflammatory molecules, which are attributed to inflammation [13] and chronic pain [14].

Our recent study suggested that LESW exerts physical energy on the inflamed bladder, which may attenuate the mitochondria-dependent apoptotic pathway and decrease bladder inflammation and overactivity in HCl-induced cystitis rat models [15].

To advance our understanding of the mechanism of LESW on CP/CPPS, we explored the effects of LESW on prostate inflammation, prostatic-to-bladder cross-sensitization, and mitochondrial dynamics regulators in a rat model of carrageenan (CAR)-induced prostatitis.

## 2. Results

### 2.1. LESW Treatment Suppressed a Decrease in the Mechanical and Thermal Thresholds Induced by CAR Injection

As shown in Figure 1a–d, intraprostatic CAR injection induced a decrease in the pain threshold, which was suppressed by LESW. Intraprostatic injections of 3% CAR and 5% CAR induced a significant decrease in the mechanical threshold 1 week after injection, and the effect of the 5% CAR injection was persistent for 2 weeks. Treatment with LESW at 100 and 300 shocks suppressed the effect of 5% CAR injection on decreasing the mechanical threshold 1 week after CAR injection. The therapeutic effect was maintained in the LESW 300 shock group 2 weeks after 5% CAR injection.

Intraprostatic injections of 3% CAR and 5% CAR induced a decrease in the thermal threshold 1 and 2 weeks after CAR injection, whose effect was suppressed by LESW at 100 and 300 shocks.

### 2.2. LESW Treatment Suppressed an Increase in the Accumulation of Inflammatory Cells in the Bladder and the Prostate

Intraprostatic 3% or 5% CAR injection induced a significant inflammatory reaction in the prostate and the bladder as evidenced by an increase in inflammatory cells. Treatment with LESW at 100 and 300 shocks significantly suppressed hyperplasia and inflammatory reaction in the prostate and the bladder (Figure 2a,b; Table 1).

### 2.3. LESW Treatment Suppressed the Expressions of Mitochondrial-Related Factors (Drp-1, MFN-2, and NLRP3) and Pain-Related Peptides (Substance P and CGRP-RCP) in the Prostate and the Bladder Induced by CAR Injection

Intraprostatic CAR injection induced a significant increase in the expressions of mitochondria-related factors Drp-1 (Figure 3a), MFN-2 (Figure 3b), and NLRP3 (Figure 4) in the stroma component of the prostate, whose effects were suppressed by LESW treatment. Intraprostatic CAR injection induced a significant increase in the expression of pain-related factors substance P (Figure 5) and CGRP-RCP (Figure 6) in the prostate and the bladder urothelium, whose effects were suppressed by LESW treatment.

### 2.4. Drp-1 and MFN-2 Gene Expression in the Prostate Tissue in CAR-Induced Prostatitis with or without LESW Treatment

Intraprostatic CAR injection induced an increase in the expressions of Drp-1 (Figure 7a) and MFN-2 (Figure 7b) in the prostate, as assessed by quantitative reverse-transcription polymerase chain reaction (RT-qPCR) analysis, whose effects were suppressed by LESW 100 shock treatment.

## 3. Discussion

This study revealed that intraprostatic CAR injection induced the downregulation of the pain threshold associated with an increase in inflammatory cell infiltration and the upregulation of the expressions of sensory peptides, such as substance P and CGRP-RCP, and mitochondrial damage-associated molecules, such as the NLRP3 inflammasome, Drp-1, and MFN-2 in the prostate tissue, all of which were also identified in the bladder and suppressed by LESW treatment (Figure 8). Our observations suggest that compromised mitochondrial integrity plays a role in prostatic inflammation, may induce hypersensitization of bladder afferents, and may produce bladder inflammation. We also postulated that LESW exerts physical energy on the inflamed prostate, which may attenuate mitochondrial-dynamics-regulators-mediated inflammation, decrease inflammatory reaction in the prostate and the bladder, and reduce pain reaction.

Several models of nonbacterial prostatitis that mimic NIH class III prostatitis have been developed [16,17]. CAR is a polysaccharide commonly used to induce inflammation and pain [18] in animal models of CP/CPPS [19]. Previous studies using CAR to induce prostatitis mainly used 3% CAR, whose models showed a 1-week response to mechanical and thermal hypersensitivity [19,20]. However, as a model to study chronic prostatitis, a 1-week duration of inflammation is insufficient; therefore, this study increased the dose to 5% CAR injection, which showed maintenance of an inflammatory reaction for 2 weeks and had a longer effect on the reduction in the mechanical threshold than 3% CAR. The therapeutic effects of morphine and LESW have been demonstrated on 1-week prostatic inflammation induced by 3% CAR or 10 mM [19,21]. The current model of 5% CAR-induced prostatitis extended the inflammatory reaction and pain response to 2 weeks. We suggest 5% CAR-induced prostatitis model is a useful model for studying CP/CPPS.

Lee et al. reported that patients with CPPS have an altered sensation of perineal pain elicited by heat, which may represent a C-fiber-mediated effect [22]. The current study showed that CAR-induced thermal hypersensitivity was reversed by LESW treatment. We suggested that a CAR-induced prostatitis model might change the somatic sensitivity.

Aizawa et al. reported that a 3% CAR-induced CP/CPPS rat model showed edema, ischemia, and inflammatory pain in the prostate, whereas no significant change was detected in bladder histology and sensation, as evaluated using a direct measurement of the mechanosensitive single-unit afferent nerve activity. They suggest that the bladder sensation is unlikely to be deteriorated in this model [20]. By contrast, our study showed that immune cell infiltration and epithelial hyperplasia are associated with the increased expressions of neuropeptide substance P and CGRP-RCP in the bladder 1 and 2 weeks after intraprostatic injection of 5% CAR; these findings were consistent with the proposal of shared afferent innervation and the potential for neural cross-talk between the prostate and the bladder [23]. We suggested that the cross-sensitization between the prostate and the bladder might be dependent on the degree of stimulation to the prostate; a stronger stimulant, such as 5% CAR, injected intraprostatically produces a stronger effect that is not seen in 3% CAR injection. Though sensory peptides were examined, these peptides are found in many types of cells and thus are not specific to nerve fibers. It has been demonstrated that substance P is mainly secreted by neurons; however, some immune cells have also been found to secrete substance P, which hints at an integral role of substance P in the neuroimmune response [24].

Mitochondrial fission and fusion are counterbalancing mechanisms acting in concert to maintain a mitochondrial network tuned to cellular function. Cellular insult and disease can lead to large rearrangements in the mitochondrial network. Indeed, dysfunction in the major components of the fission and fusion machineries of the mitochondria, including dynamin-related protein 1 and mitofusin 1 and 2 (MFN1 and MFN2), was found to result in neurodegenerative diseases [25]. In addition, altered mitochondrial dynamics are seen commonly in diseases as triggers for inflammation [26].

Mitochondrial dysfunction has been suggested to be an important pathological mechanism in chronic inflammatory and urogenital diseases [14,27]. The present study found that intraprostatic CAR injection induced an upregulation of the expressions of Drp-1, MFN2, and NLRP3, all of which were downregulated after LESW treatment. An increase in mitochondrial fission via the regulation of Drp-1 enhanced the generation of ROS [22], which ensued adverse reactions, such as inflammation and cell senescence [13]. Mitochondria are reported to be involved in inflammasome activation through ROS production or interaction with the components of the NLRP3 inflammasome. The NLRP3 inflammasome is activated by diverse stimuli, and multiple molecular and cellular events have been linked to the pathogeneses of several inflammatory disorders [28].

Scaini et al. revealed dysregulation of mitochondrial dynamics, manifested as elevated levels of Mfn-2, short Opa-1, and Fis-1, and dysregulation of neuroinflammatory pathways in the pathophysiology of major depressive disorder [29]. Furthermore, elevated MFN-2 levels were found to be involved in metabolic changes and inflammation in chondrocytes and bone joints in an aging rat model [30]. Our current data showed that LESW can reduce the expressions of Drp-1, MFN-2, and NLRP3 induced by intraprostatic CAR injection, and results suggested that LESW, which regulates mitochondrial dynamics regulators and neuroinflammation, might have anti-inflammatory and analgesic effects on CP/CPPS. Further study with other mitochondrial-related molecules and functional assays may yield more information as to the benefits of shock wave therapy in this animal model.

Results of the RT-qPCR analysis showed that CAR induced an increase in the expressions of Drp-1 and MFN-2 at various levels. LESW 100 shock treatment downregulated the expressions of Drp-1 and MFN-2, whose effects were not seen in the 300 shock-treated group. These findings were inconsistent with the results of the IHC study. Post-transcriptional modification has been linked to translational efficiency and expression of the protein content in tissues [31,32]. We proposed that LESW may influence post-transcriptional modification and cause the downregulation of the protein expressions of Drp-1 and MFN-2 in the inflamed prostate. In addition, the gene expressions between control and 5% CAR-treated prostates in Drp-1 and MFN-2 showed no significant difference. Further study with a larger sample size may be necessary to elucidate the real role of Drp-1 and MFN-2 in a CAR-induced prostatitis model in rats.

This study had some limitations. Although CAR-induced prostatitis model revealed that mitochondrial factors were positively associated with inflammatory pain, a gap remains between the current prostatitis rat models and the actual CP/CPPS in humans. Thus, the therapeutic effects of shock waves on the modulation of mitochondria-related factors linked to chronic pain require more evidence, such as a larger number of animals in each group, changes in inflammatory molecules, and human data, to confirm their true value.

Prostatodynia is the main symptom of chronic prostatitis and one of the major reasons that patients go to the hospital for treatment. Mechanical allodynia, assessed by using the von Frey filament test or heat stimulation threshold, was widely used in the animal models of chronic prostatitis [19,33]. Most research on CAR-induced prostatitis models in rats evaluated the pain behavior at 1 week. The current study extended the observation from 1 week to 2 weeks and found that 5% CAR-induced pain and inflammation were significant at 1 week; however, the degree of pain and inflammation was decreased at 2 weeks. A longer study with more extensive sampling will be needed to clarify the plateau and return to baseline after intraprostatic CAR injection.

Recent publications have demonstrated that the proportions of effector T-cell subpopulations, particularly central memory T cells, T helper (Th)1, Th17 and Th22 cells, and CD4^+^ lymphocytes, in the peripheral blood of CP/CPPS patients were significantly increased compared with those of healthy controls [34,35]. Further research on abnormal differentiation of T lymphocytes might advance our understanding of the pathogenesis of CP/CPPS and impact of LESW treatment.

In conclusion, our study found that LESW attenuation of mitochondrial dysregulation inhibits the expression of sensory peptides, prostate inflammatory reactions, and pain responses in a rat prostatitis model. These findings support the application of LESW for the treatment of CP/CPPS.

## 4. Materials and Methods

### 4.1. Animals

Adult male Sprague–Dawley rats weighing 300–350 g were used in this study. Animals were kept under constant temperature and humidity in a 12 h dark–light cycle.

This study was approved by the Institutional Animal Care and Use Committee of Chang Gung Memorial Hospital (IACUC no. 2019121814) and complied with the NIH guidelines for the care and use of laboratory animals. At the end of the experiments, the animals were deeply anesthetized by intramuscular injection of Zoletil 50 (25–50 mg/kg) and Xylazine (10–23 mg/kg) for humane sacrifice, followed by transcardiac perfusion with Krebs buffer.

### 4.2. CAR-Induced Prostatitis Model

All experimental animals were randomly divided into five groups (N = 6 for each group): sham control group, 3% CAR-induced prostatitis group, 5% CAR-induced prostatitis group, and 5% CAR-induced prostatitis groups with LESW (100 or 300 pulses; 0.12 mJ/mm^2^; frequency of two pulses per second). The experimental animals were anesthetized with isoflurane (5% during induction and 2% during maintenance), and a small incision was made in the midline of the lower abdomen following a sterile technique. The bladder and the prostate were carefully exposed from the surrounding tissues. CAR (Sigma, MO, USA) was dissolved in 50 μL of sterile saline at concentrations of 3% or 5% and injected into the left and right ventral lobes of the prostate with a 30-gauge needle. The wound was closed in layers, and an antibacterial cream was applied to the wound. The sham control group received 50 μL of sterile saline injection.

### 4.3. LESW Treatment

Animals received LESW treatment at 24 h, 7 days, and 8 days after 5% CAR injection. The current shock wave treatment was modified from a previous report [21,36]. The shock wave probe (SD-1, Storz, Tägerwilen, Germany) was gently placed over the skin surface above the prostate area after the application of an ultrasound transmission gel.

### 4.4. Assessment of Pain Behaviors

Pain behaviors were evaluated by blind observers on the day before surgery and at 1 week and 2 weeks after saline or CAR injection. Measurements were repeated five times for each rat with a 5 min rest period. The experimental animals were acclimated in individual observation boxes for 30 min before testing. The mechanical pain threshold was tested at the base of the scrotum using a von Frey filament attached to a dynamic plantar aesthesiometer (Ugo Basile, Comerio, Italy) [19]. The skin was placed on a wire mesh between the penis and scrotum and stimulated with von Frey filaments with gradually increasing pressure until the animal left its original position.

Radiant heat from a plantar analgesia test device (Ugo Basile, Comerio, Italy) was irradiated on the scrotal skin. The heat stimulation threshold was defined as the duration from the start of the test to the location where the rats moved.

### 4.5. Immunohistochemical (IHC) Assay

The prostate and the bladder were harvested 14 days after saline or CAR injection, fixed in 10% formaldehyde buffer for 24–48 h, and then embedded in paraffin for H&E staining and immunohistochemistry.

The prostate and the bladder tissue sections for H&E staining and immunohistochemistry were cut into 3 μm sections, de-paraffinized in xylene for 10 min, and rehydrated with gradient descent ethanol at room temperature. Sections were stained in Gill II hematoxylin solution (Leica, Wetzlar, Germany, 3801522) for 1 min. Counterstaining was conducted in Eosin Y solution (Leica, 3801602) for 5 min. The xylene-based mounting agent was found to seal the coverslip. The inflammatory reaction of CAR-induced cystitis was graded on a score of 0–3 as follows: 0, no evidence of inflammatory infiltration or interstitial edema; 1, mild (few inflammatory cell infiltrates and little or no interstitial edema); 2, moderate (moderate amount of inflammatory cell infiltrates and moderate interstitial edema); 3, severe (diffuse presence of large amounts of inflammatory cell infiltrates and severe interstitial edema) [37,38].

Moreover, 3% hydrogen peroxide was used for 10 min to block endogenous peroxidase activity. Tissue sections were incubated with anti-Drp-1 (1:2000 dilution; Abcam, Cambridge, UK), anti-mitofusin 2 (MFN-2) (1:25 dilution; Abcam), anti-NACHT, LRR, and PYD domains-containing protein 3 (NLRP3) (1:200 dilution; Invitrogen, Waltham, MA, USA), anti-substance P (1:200 dilution; Invitrogen), and anti-CGRP-receptor component protein (CGRP-RCP) (1:200 dilution; Abcam) at 37 °C for 60 min, and then washed twice with PBS buffer. Then, we applied the Primary Antibody Amplifier Quanto (Waltham, MA, USA) and incubated for 10 min. After that, we applied the HRP Polymer Quanto (Waltham, MA, USA) and incubated for 10 min. After washing in PBS, we added 30 µL of DAB Quanto Chromogen (Waltham, MA, USA) to 1 mL DAB Quanto Substrate (Waltham), mixed, and applied to the tissue. We incubated for 5 min. Then, we used a xylene-based mounting agent for coverslip sealing. IHC staining scores were calculated based on the proportion of positively stained areas in tissue section images. Three randomly selected fields were analyzed using Image-Pro Plus version 6.0 software (Media Cybernetics, Inc., Rockville, MD, USA).

### 4.6. Total RNA Extraction and RT-qPCR Analysis

The RNA was extracted using RNA Isolater Total RNA Extraction Reagent (Vazyme, Nanjing, China). The concentration was quantified using an Epoch spectrophotometer with a Take3 plate (BioTek, Winooski, VT, USA). The resulting cDNA was amplified using the specific 20X gene expression assays for SYBR green of β-actin, Drp-1, and MFN-2 for rats (#600119 Topgen Biotech., Kaohsiung, Taiwan) (primers listed in Table 2). qPCRs were performed with ChamQ Universal SYBR qPCR Master Mix (Vazyme) in a 10 µL reaction on the StepOne Plus Real-Time PCR System (Applied Biosystems, Waltham, MA, USA) in accordance with the manufacturer’s instructions. Then, 10 ng of cDNA was amplified in triplicate with appropriate non-template controls. Amplification data were normalized to β-actin expression. Quantification of relative expression (reported as arbitrary units (a.u.)) was performed using the 2^−∆∆Ct^ relative quantification method. qPCR data showed a variability coefficient of Ct always >2% of the mean values.

### 4.7. Statistical Analysis

Experimental data are presented as means ± standard deviations. Statistical analyses were performed using a one-way analysis of variance with the Tukey post-test. A *p*-value of 0.05 was considered significant. All statistical operations were performed using SPSS Statistics for Windows, version 14.0 (SPSS, Inc., Chicago, IL, USA).

## Figures and Tables

**Figure 1 ijms-24-03898-f001:**
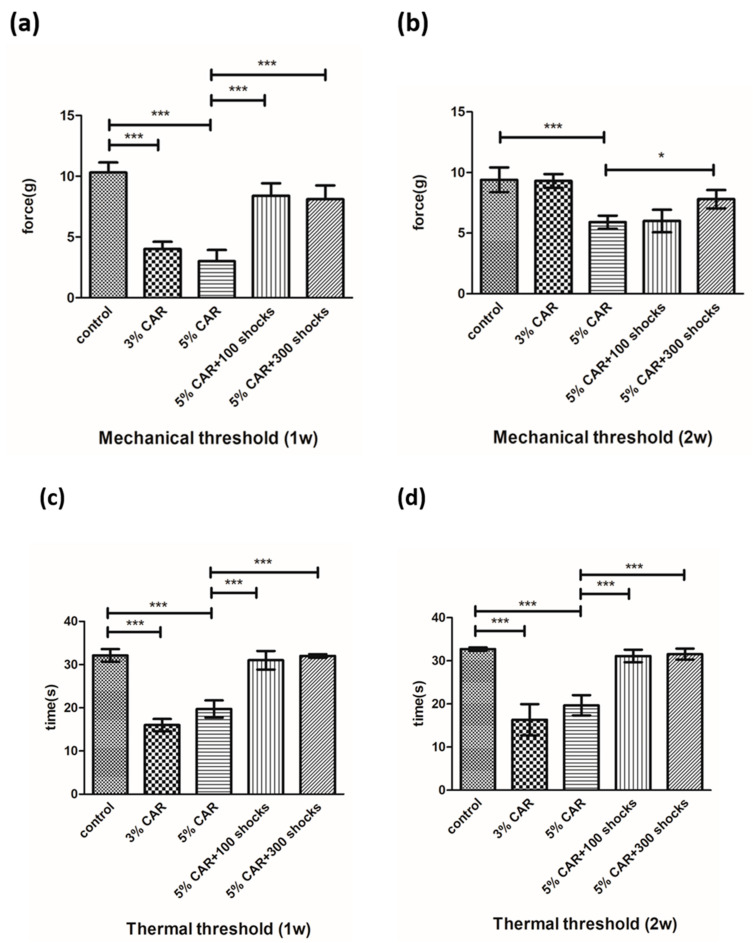
Effects of CAR and LESW treatment on mechanical stimulation and heat stimulation (*n* = 6 rats per group): (**a**) mechanical threshold 1 week after CAR injection, (**b**) mechanical threshold 2 weeks after CAR injection, (**c**) thermal threshold 1 week after CAR injection, (**d**) thermal threshold 2 weeks after CAR injection (* *p* < 0.05, *** *p* < 0.001).

**Figure 2 ijms-24-03898-f002:**
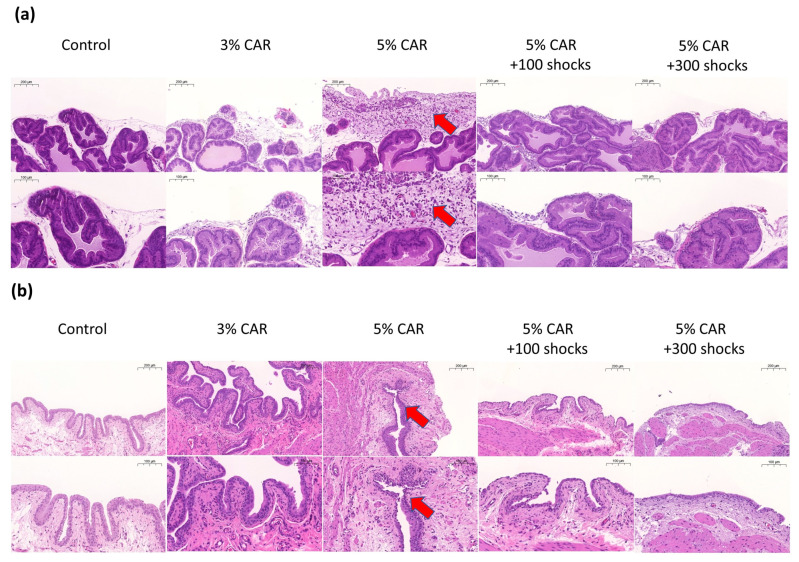
Hematoxylin and eosin (H&E) staining of carrageenan-induced prostatitis with or without LESW treatment (*n* = 6 rats per group). (**a**) H&E staining of the prostate and (**b**) H&E staining of the bladder. Magnification of the upper and lower panels ×50 and ×100, respectively. The arrows show uroepithelium hyperplasia and inflammatory cell infiltration.

**Figure 3 ijms-24-03898-f003:**
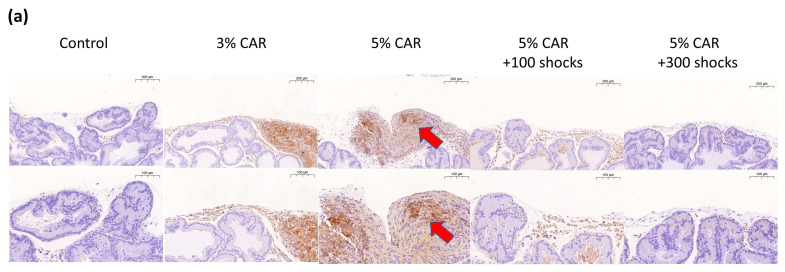
Drp-1 and MFN-2 expressions in CAR-induced prostatitis with or without LESW treatment. (**a**,**c**) Intraprostatic CAR injection induced a significant increase in Drp-1 expression in the stromal component of the prostate, whose effects were suppressed by LESW treatment. (**b**,**d**) Intraprostatic CAR injection induced a significant increase in MFN2 expression in the stromal component of the prostate, whose effects were suppressed by LESW treatment. Magnification of the upper and lower panels ×50 and ×100, respectively. The arrows indicate the IHC-positive regions (** *p* < 0.01, *** *p* < 0.001).

**Figure 4 ijms-24-03898-f004:**
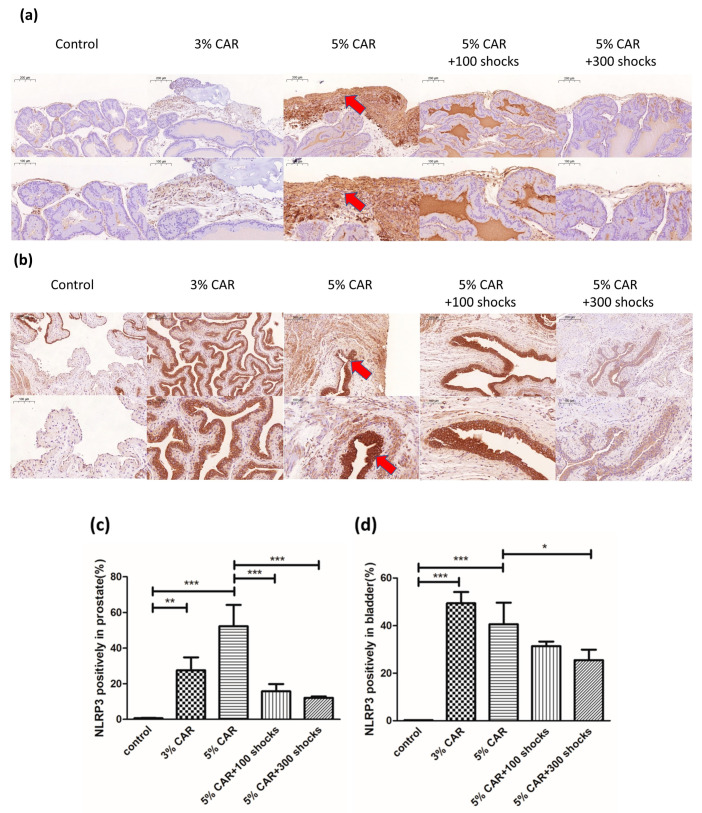
NLRP3 expression in CAR-induced prostatitis with or without LESW treatment. (**a**,**c**) Intraprostatic CAR injection induced a significant increase in the expression of NLRP3 in the stromal component of the prostate (**a**,**c**) and the bladder urothelium (**b**,**d**), whose effects were suppressed by LESW treatment. Magnification of the upper and lower panels ×50 and ×100, respectively. The arrows indicate the IHC-positive regions (* *p* < 0.05, ** *p* < 0.01, *** *p* < 0.001).

**Figure 5 ijms-24-03898-f005:**
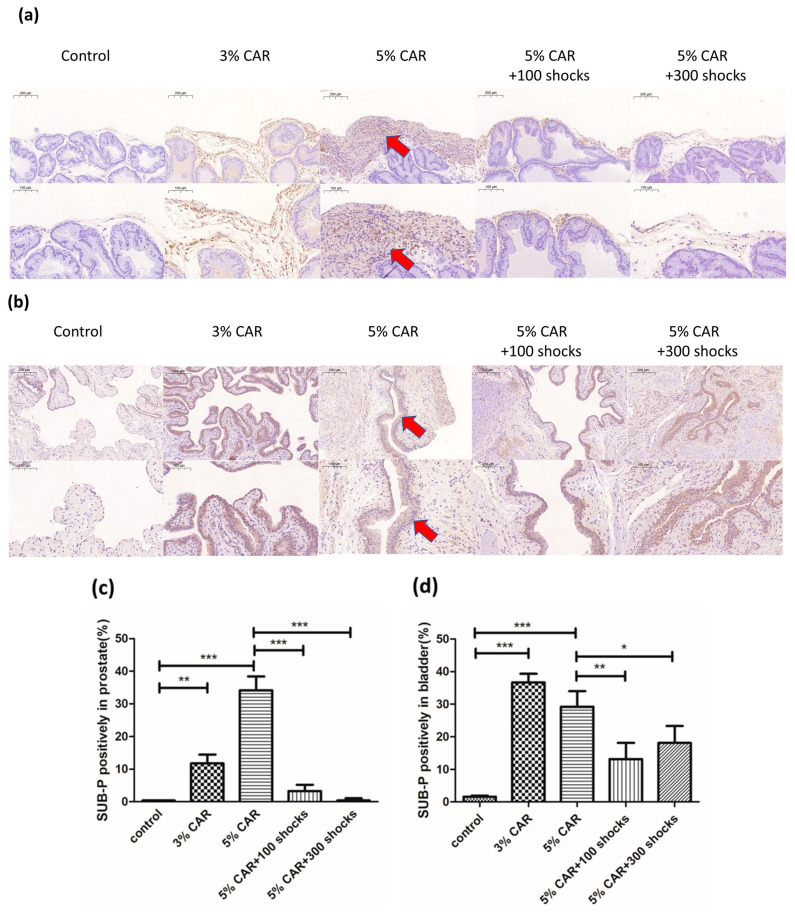
Substance P expression in CAR-induced prostatitis with or without LESW treatment. (**a**,**c**) Intraprostatic CAR injection induced a significant increase in the expression of substance P in the stromal component of the prostate and the bladder uroepithelium (**b**,**d**), whose effects were suppressed by LESW treatment. Magnification of the upper and lower panels ×50 and ×100, respectively. The arrows indicate the IHC-positive regions (* *p* < 0.05, ** *p* < 0.01, *** *p* < 0.001).

**Figure 6 ijms-24-03898-f006:**
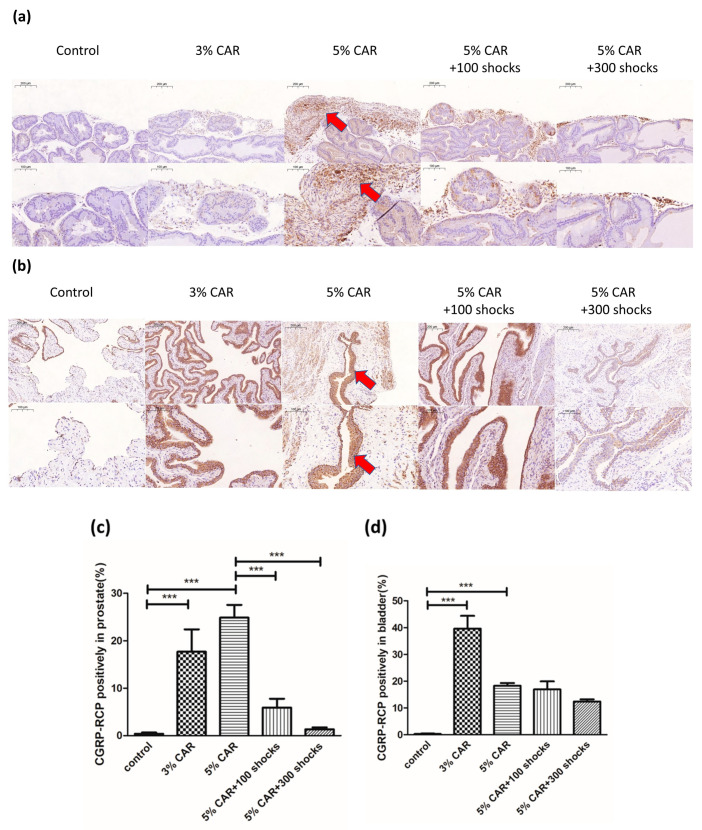
CGRP-RCP expression in CAR-induced prostatitis with or without LESW treatment. (**a**,**c**) Intraprostatic CAR injection induced a significant increase in the expression of CGRP-RCP in the stromal component of the prostate, whose effect was suppressed by LESW treatment. (**b**,**d**) Intraprostatic CAR injection induced a significant increase in the expression of CGRP-RCP in the bladder uroepithelium, whose effect was not altered by LESW treatment. Magnification of the upper and lower panels ×50 and ×100, respectively. The arrows indicate the IHC-positive regions (*** *p* < 0.001).

**Figure 7 ijms-24-03898-f007:**
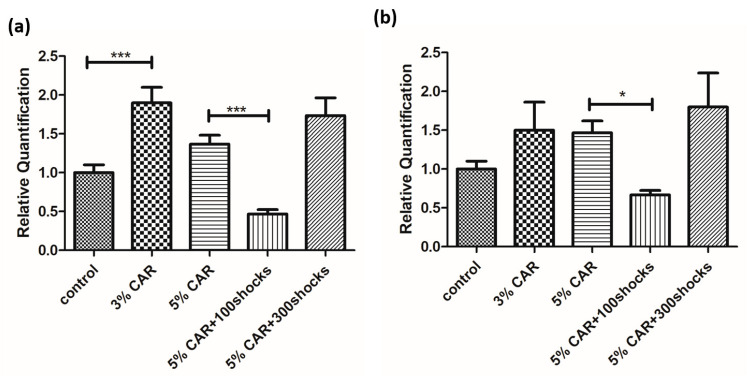
RT-qPCR analysis of the *prostate tissue* in CAR-induced prostatitis with or without LESW treatment. (**a**) Drp-1 gene expression in each group, and (**b**) MFN-2 gene expression in each group (* *p* < 0.05, *** *p* < 0.001).

**Figure 8 ijms-24-03898-f008:**
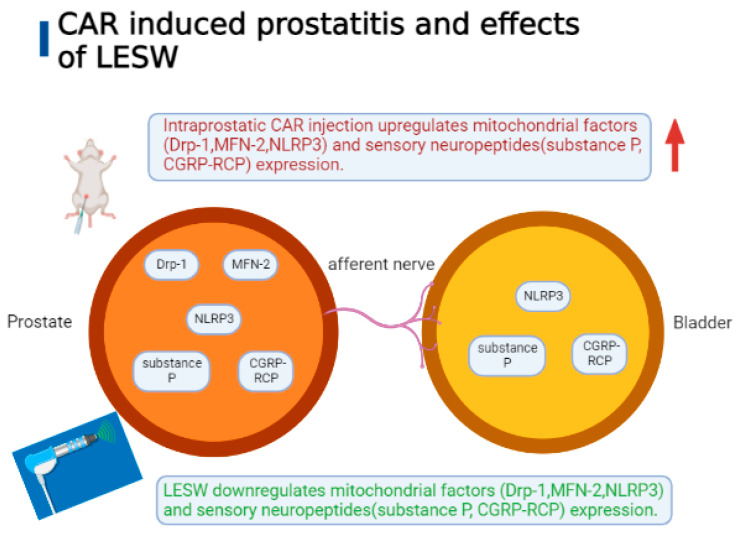
Intraprostatic CAR injection induced neuroinflammation and upregulation of Drp-1, MFN-2, NLRP3, substance P, CGRP-RCP, afferent innervation, and potential neural crosstalk to the prostate and the bladder, effects that could be reduced by LESW treatment. LESW might partially reverse mitochondrial dynamics dysregulation and neuroinflammation by modulating mitochondria-related factors and sensory neuropeptides.

**Table 1 ijms-24-03898-t001:** On day 8, effects of control, 3% CAR, 5% CAR, 5% CAR + 100 shocks, and 5% CAR + 300 shocks on inflammation (inflammatory cell scoring and edema scoring).

Prostate	Edema	Inflammatory Cell
A: Control	1.33 ± 0.52	1.33 ± 0.52
B: 3% CAR	2.33 ± 0.52	1.67 ± 0.52
C: 5% CAR	2.67 ± 0.52	2.67 ± 0.52
D: 5% CAR + 100 shocks	1.33 ± 0.52	1.33 ± 0.52
E: 5% CAR + 300 shocks	1.33 ± 0.52	1.33 ± 0.52
** *p* ** **-value**		
A vs. B	0.02	0.796
A vs. C	0.001	0.001
C vs. D	0.001	0.001
C vs. E	0.001	0.001
Data presented as means ± SD		
N = 6 for each group		
**Bladder**	**Edema**	**Inflammatory Cell**
A: Control	1.33 ± 0.58	1.33 ± 0.58
B: 3% CAR	2.33 ± 0.58	2.67 ± 0.58
C: 5% CAR	2.33 ± 0.58	2.67 ± 0.58
D: 5% CAR + 100 shocks	1.33 ± 0.58	1.67 ± 0.58
E: 5% CAR + 300 shocks	1.33 ± 0.58	1.33 ± 0.58
** *p* ** **-value**		
A vs. B	0.02	0.001
A vs. C	0.02	0.001
C vs. D	0.02	0.02
C vs. E	0.02	0.001
Data presented as means ± SD		
N = 6 for each group		

**Table 2 ijms-24-03898-t002:** Primer sequence of the selected candidate reference genes.

Gene Primer	Sequence (5’-3’)
β-actin-F	AGGCCCCTCTGAACCCTAAG
β-actin-R	CAGCCTGGATGGCTACGTACA
MFN2-F	AGCACTTTGTCACTGCCAAGAAA
MFN2-R	GGACCTGCTCTTCTGTAGTAACTGG
Drp1-F	TGAAGGAACGGCAAAGTACATTG
Drp1-R	GGATCAACAGATTCTAAGGTTCGC

## Data Availability

The data presented in this study are available in the article.

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
