# Peer review of "Low-Energy Shock Wave Suppresses Prostatic Pain and Inflammation by Modulating Mitochondrial Dynamics Regulators on a Carrageenan-Induced Prostatitis Model in Rats"

_ijms, 2023, doi:10.3390/ijms24043898_

Round 1
Reviewer 1 Report (Previous Reviewer 3)
The author responded well to my comments and the current version is ready for publication.
Reviewer 2 Report (Previous Reviewer 2)
As previously said I like the manuscript and I also believe this new version is emproved.
However, I still don't understand why Methods are at the end of the discussion and not after Introduction as should be.
Reviewer 3 Report (New Reviewer)
It has long been pointed out that the concept of CP/CPPS disease and its treatment are very difficult.
In this study, we examined the effectiveness of low dose shock waves in CP/CPPS rats, and the content of the mechanism is also in-depth.
I think that the content of this paper is very meaningful.
It would be great if you could provide more detail on the therapeutic prospects for CP/CPPS in humans.
This manuscript is a resubmission of an earlier submission. The following is a list of the peer review reports and author responses from that submission.
Round 1
Reviewer 1 Report
This study which aims to show that low energy shock wave suppresses pain and inflammation is topical and of potential interest in terms of mechanisms underlying prostatitis. However there are some concerns that decrease enthusiasm. For example, though the authors show changes in allodynic behavior (a change in mechanical threshold is not necessarily 'pain' response), it is not clear when the changes peak/plateau and return to baseline. thus a more extensive sampling will be needed. is there any changes in somatic sensitivity? There was no mention as to whom did the scoring for the inflammatory infiltrates and also the quality of the histology shown could be improved.
Though sensory peptides were examined, it should be discussed that these peptides are found in many types of cells thus are not specific to nerve fibers. it is also not clear how the authors link changes to just a few general proteins that are linked to mitochondrial fission/fusion (and one linked to the inflammasome platform)- that these were described as only playing a role in mitochondrial dysfunction and apoptosis. Other targets may yield more information as to the benefits of shock wave therapy in this animal model.
Reviewer 2 Report
I read with interest this manuscript and I commend the authors. This is a very interesting study in my opinion.
However, I found some issues that must be edited.
1- Methods session cannot be at the end of the manuscript. It must be after the introduction and before the results. Besides that, methods are well described and clear to read.
2 - Results are well written and clear
3- I would implement the limitations: only 6 mice for group, so a number limitation. Furthermore, the pain evaluation in mice cannot be as precise as in humans so might not be reproducible.
Reviewer 3 Report
Low-energy shock wave has potential to be a novel treatment for chronic prostatitis/chronic pelvic pain syndrome (CP/CPPS). However, its mechanism of treatment remains unclear. The authors focused on the pathological changes of rat’s prostate and bladder after treatment. There are some major questions should be concerned:
1. The references between CP/CPPS and LESW should be provided in the introduction part, such as the LESW impact the metabolisms for CP/CPPS patients (PMID: 35911714), and the application of LESW in patients (PMID: 34585431; PMID: 35671994).
2. In Figure2, group (3% CAR) showed significant inflammatory cell infiltration and uroepithelium hyperplasia. Why not set up treatment groups for 3% CAR;
3. In Figure 7, the difference of expressions between control and 5%CAR in Drp-1 and MFN-2 showed no significant difference. So, these genes may be not strongly associated with therapeutic implications
4. Have the authors evaluated the change proportion of Th1 or Th17 cells in peripheral blood or spleen, cause several articles reported that Th1 and Th17 proportion inflected the severe of CP/CPPS, as mentioned in PMID: 33124198, 35693826.
5. In material and method, references for LESW energy settings should be added.
6. There are several letters in uppercase on line 209.
